



# 1 Performance of three machine learning algorithms for predicting soil

# 2 organic carbon in German agricultural soil

Ali Sakhaee[1], Anika Gebauer[2], Mareike Ließ[2], Axel Don[1]
[1]Thünen Institute of Climate Smart Agriculture, Braunschweig, Germany
[2]Department Soil System Science, Helmholtz Centre for Environmental Research – UFZ, Halle (Saale), Germany
*Correspondence to*: Ali Sakhaee (a.sakhaee@thuenen.de)
**Abstract**
Soil organic carbon (SOC), as the largest terrestrial carbon pool, has the potential to influence climate change and
mitigation, and consequently SOC monitoring is important in the frameworks of different international treaties.
There is therefore a need for high resolution SOC maps. Machine learning (ML) offers new opportunities to do
this due to its capability for data mining of large datasets. The aim of this study, therefore, was to test three
commonly used algorithms in digital soil mapping – random forest (RF), boosted regression trees (BRT) and
support vector machine for regression (SVR) – on the first German Agricultural Soil Inventory to model
agricultural topsoil SOC content. Nested cross-validation was implemented for model evaluation and parameter
tuning. Moreover, grid search and differential evolution algorithm were applied to ensure that each algorithm was
tuned and optimised suitably. The SOC content of the German Agricultural Soil Inventory was highly variable,
ranging from 4 g kg⁻¹ to 480 g kg⁻¹. However, only 4% of all soils contained more than 87 g kg⁻¹ SOC and were
considered organic or degraded organic soils. The results show that SVR provided the best performance with
RMSE of 32 g kg⁻¹ when the algorithms were trained on the full dataset. However, the average RMSE of all
algorithms decreased by 34% when mineral and organic soils were modelled separately, with the best result from
SVR with RMSE of 21 g kg⁻¹. Model performance is often limited by the size and quality of the available soil
dataset for calibration and validation. Therefore, the impact of enlarging the training data was tested by including
1223 data points from the European Land Use/Land Cover Area Frame Survey for agricultural sites in Germany.
The model performance was enhanced for maximum 1% for mineral soils and 2% for organic soils. Despite the
capability of machine learning algorithms in general, and particularly SVR, in modelling SOC on a national scale,
the study showed that the most important to improve the model performance was separate modelling of mineral
and organic soils.
**1 Introduction**
Soil organic carbon (SOC) is the largest terrestrial carbon pool (Wang et al., 2020) and plays an essential role in
agriculture. Since SOC influences various physical, chemical and biological properties of soil (Reeves, 1997),
numerous studies recognise it as a crucial indicator of soil quality (Castaldi et al., 2019; Meersmans et al., 2012;
Reeves, 1997). Thus, its decline is identified as a threat that leads to soil degradation (Castaldi et al., 2019; Poeplau
et al., 2020). Moreover, when considering carbon sequestration, the SOC pool provides the option for climate
change mitigation (Meersmans et al., 2012; Ward et al., 2019). Consequently, SOC monitoring is important in the
frameworks of various international treaties such as the European Union Soil Thematic Strategy and the United
Nations Framework Convention on Climate Change (Meersmans et al., 2012; Poeplau et al., 2020). There is
therefore growing interest in understanding the spatial distribution of SOC at different scales in response to



increasing demand for a better assessment of SOC (Minasny et al., 2013). This is particularly important for
agricultural land due to its potential for carbon sequestration (Lal, 2004).
In digital soil mapping (DSM), a soil attribute is formulated as an empirical quantitative function of seven factors:
soil properties, climate, organism, topography, parent material, time and spatial position (McBratney et al., 2003).
Therefore, this function, known as the SCORPAN model, can be applied to spatially predict the soil attribute of
interest (Minasny et al., 2013). Within this framework, machine learning algorithms aim to automatically extract
the information from the data for predictive purposes (Behrens et al., 2005). This is particularly intriguing in view
of the expansion of databases at a different scale in soil science and the complexity of the covariates in recent years
(McBratney et al., 2003; Wadoux et al., 2020), thus making DSM cost-effective, time-efficient and applicable over
large areas with good results (Behrens and Scholten, 2006; Camera et al., 2017).
Despite the advantages of DSM, it is crucial to consider that its application requires soil databases of an adequate
sample size for training and testing. Furthermore, consistent and quality-checked datasets are a prerequisite for
DSM. Several soil inventories and monitoring networks for SOC have been formed on a national scale in countries
such as Sweden (Poeplau et al., 2015), France (Belon et al., 2012; Meersmans et al., 2012), Denmark (Taghizadeh-
Toosi et al., 2014) and Scotland (Chapman et al., 2013). Nonetheless, the most critical shortcomings of soil
inventories in Germany are the lack of a large-scale, high-quality SOC inventory (Wiesmeier et al., 2012) with a
periodic and standardised sampling focus on agricultural soils (Prechtel et al., 2009). These issues have now been
solved in the first German Agricultural Soil Inventory ( Poeplau et al., 2020). This inventory was conducted on a
national scale with a sampling depth down to 100 cm at 3104 sampling sites covering agricultural land.
Furthermore, on a European scale, the Land Use/Land Cover Area Frame Survey (LUCAS) undertaken in 2009 is
the first harmonised topsoil survey with physico-chemical analyses of georeferenced topsoil samples in 23
European states (Tóth et al., 2013). Therefore, by taking advantage of DSM and both the German Agricultural Soil
Inventory and the LUCAS survey, it is possible to regionalise single-point measurements to obtain complete high-
resolution cover soil data and thus provide a baseline for SOC monitoring as well as for environmental and climatic
modelling for Germany.
Boosted regression trees (BRT), random forest (RF) and support vector machine for regression (SVR) are among
the most used algorithms in DSM (Padarian et al., 2020). For example, Martin et al. (2014) predicted topsoil SOC
on a national scale for France using the BRT algorithm and compared its results when the same algorithm was
coupled with a geostatistical approach. They concluded that since spatial autocorrelation is not feasible in most
national inventories, the BRT algorithm alone is sufficient for this purpose. This algorithm was also used on a
national scale in China for data from the 1980s and 2010s in order to predict topsoil SOC and its spatial-temporal
change, as well as the main drivers of its variability (Wang et al., 2021). Moreover, RF has become more popular
in DSM due to its relative simplicity and performance. For example, this algorithm was implemented to map
topsoil SOC on a national scale in Madagascar and obtain its main drivers (Ramifehiarivo et al., 2017).
Ramifehiarivo et al. (2017)concluded that that the uncertainty of the algorithm was lower when compared with the
maps formerly generated for the country. Moreover, this algorithm was compared with the Cubist model for
mapping SOC at different resolutions on a regional scale in China and could outperformed it (Li et al., 2021).
Fewer studies have used SVR to predict SOC than RF. Studies have mainly implemented SVR on a regional scale
with a limited number of samples (Forkuor et al., 2017; Were et al., 2015) or on a national scale (Switzerland)
with very few samples (150 samples from the European LUCAS survey) (Zhou et al., 2021). However, in a study





comparing different algorithms, including SVR and RF, on a continental scale and within each country in Latin
America, the results indicated that the best-performing algorithm varied in different countries (Guevara et al.,
2018). This difference mainly depended on sample dispersity and also country size which affects the heterogeneity
of land use and environmental conditions.
Another important consideration when applying machine learning is the impact of the parameter-tuning strategy
in algorithm performance. This is particularly crucial when the objective of the study is the comparisons of
different machine learning algorithms. Although some algorithms are less sensitive to tuning, this step is more
important for others, particularly those with a higher number of parameters (Tziachris et al., 2020; Wadoux et al.,
2020). Furthermore, as algorithms differ by the type of their parameters, continuous or discrete, the chosen strategy
should be in accordance with this difference. This is particularly more important for algorithms with continuous
parameters.  For example, it has been shown that that the performance of SVR and BRT is better and more stable
when optimised by differential evolution (DE) algorithm than tuned by grid search (Zhang et al., 2011; Gebauer
et al., 2020). Despite this importance, in a review of studies that have applied DSM, Wadoux et al. (2020) state
that almost half of them implemented parameter tuning, with grid search the most common strategy for this
purpose. This finding indicates that the role of parameter tuning and optimisation is unfortunately undermined in
DSM. This is particularly evident when the application of machine learning in this field is compared with other
fields, where various studies have shown the impact of parameter-tuning strategies on the performance of
algorithms such as SVR and BRT (Liang et al., 2011; Santos et al., 2021; Bhadra et al., 2012; Deng et al., 2019).
The aim of the present study was therefore i) to address the above-mentioned parameter-tuning issue and
consequently provide a true comparison of the performance of BRT, RF, and SVR in modelling the SOC contents
of German agricultural topsoil (0-30 cm), ii) to assess the impact of training data size by extending the data of the
German Agricultural Soil Inventory with LUCAS data for model calibration, and iii) to develop a two-model
approach to address the high variability of SOC in German agricultural soils and compare it with a single-model
approach.

## 2 Materials and methods

### 2.1 Soil data

The models were built using SOC content data from two soil inventories. The first dataset was from the German
Agricultural Soil Inventory, which consists of 3104 sites with a fixed grid of 8x8 km throughout Germany (Poeplau
et al., 2020). The sites were sampled and analysed for different soil properties, including SOC content measured
via dry combustion, for the upper 30 cm of the soil between 2012 and 2018. The second dataset was the European
LUCAS survey that provides SOC content, similarly measured via dry combustion, for all EU countries, with the
sampling depth limited to 0-20 cm (Tóth et al., 2013). Therefore, in order to harmonise the depths of both datasets,
these were subdivided into mineral and organic soil classes according to a SOC threshold value of 87.0 g kg$^{-1}$
considering all soils above this threshold as organic soils comprising peat soils and disturbed and degraded peat
soils (Poeplau et al., 2020). Linear correlation functions between 0-30 cm and 0-20 cm were derived for each soil
class of the German Agricultural Soil Inventory separately. These functions were applied to the corresponding soil
class from the LUCAS data in order to estimate 0-30 cm topsoil SOC. With a slope of 0.881 for mineral soils and
1.02 for organic soils, they changed the mean of LUCAS data by less than 6%. The depth-extrapolated values of



mineral and organic soils were then combined to form the complete dataset. The 0-30 cm LUCAS data generated
and the original 0-20 cm LUCAS data were then used by each algorithm to check the effect of depth extrapolation.
**2.2 Covariates**
Covariates from multiple sources were included to approximate the SCORPAN factors throughout Germany. In
case of multiple data products for one covariate, the one with the best quality (least artefacts), and the highest
spatial resolution was added. These were then resampled in ArcGIS (ESRI, 2013) using the INSPIRE standard
grid at 100 m resolution (Eurostat grid generation tool for ArcGIS). The resampling method was either the nearest
neighbour for categorical covariates or bilinear interpolation for continuous covariates. The same INSPIRE grid
was used to rasterise the vector covariates as well. Finally, they were stacked and overlaid on SOC databases in
order to extract the values of sampling points.
Following the SCORPAN framework, 24 covariates including x and y spatial positions were compiled. In order to
capture climate factors (C factor), precipitation (DWD, 2018c), sunshine duration (DWD, 2017), summer days
(DWD, 2018b) and minimum temperature (DWD, 2018a) were used according to the study of Schneider et al.
(2021). Using principal component analysis, these four covariates were indicated to be the most important among
34 available climate factors for SOC in the German Agricultural Soil Inventory dataset. Moreover, land use is one
of the main drivers of SOC variability at a national scale (Poeplau et al., 2020). Thus, the land-use map from the
official topographic information system (BKG, 2019) with its corresponding classes according to the German
Agricultural Soil Inventory was rasterised and included. This is a categorical covariate, representing the organism
factor of SCORPAN (O factor), that distinguishes croplands from grasslands and captures their spatial distribution
throughout Germany.
The European Digital Elevation Model (EUDEM) (European Union Copernicus Land Monitoring Service, 2016)
and six covariates derived from this layer were also added to integrate the topography and relief parameters (R
factor). Slope, plan curvature and profile curvature, generated on SAGA (Conrad et al., 2015), were included to
capture the slope's gradient, convexity-concavity and convergence-divergence. These factors influence the soil
distribution throughout the landscape, e.g. affecting flow over the surface, thus impacting SOC and its dynamic
(Ritchie et al., 2007). Moreover, north-south and east-west aspects were obtained from EUDEM as these influence
soil development and subsequently affect SOC (Carter and Ciolkosz, 1991). The Topographic Wetness Index
(TWI), generated on SAGA (Conrad et al., 2015), was also added since it captures the soil moisture distribution
of the landscape and has a direct correlation with SOC (Pei et al., 2010). A geomorphographic map of Germany
containing 25 geomorphic categories was also used to distinguish between four different landscape areas of the
country: North German lowlands, highlands, Alpine foothills and the Alps.
Continuing with the framework, a large-scale soil landscape unit map ("Bodengrosslandschaft") (Richter et al.,
2007) comprising 38 classes was used. This covariate divides Germany by various geo-factors that can be compiled
into a map with 12 soil regions representing mainly the parent materials. Similarly, large-scale soil-climate region
map ("Bodenklima") (Roßberg, 2007) with 50 classes was added. Moreover, Germany's hydrogeological unit map
(BGR and SGD, 2019) provides information about lithology and its hydrological characteristics. These categorical
maps were rasterised and applied to the model as the P factor of SCORPAN. Moreover, the soil factor of the
framework (S factor) was captured by eight covariates that represent different aspects of its properties: the map of
organic soils (Roßkopf et al., 2015) that distinguishes mineral soils from organic ones and explains their spatial



distribution throughout the country, as well as the map of nitrogen (Ballabio et al., 2019) and clay content (Ballabio
et al., 2016) since they directly correlate with SOC. As nitrogen is a crucial component of soil organic matter,
regions with higher total nitrogen have higher SOC (Ballabio et al., 2019). Also for clay content, different studies
have shown that coarser soil textures tend to have a lower accumulation of SOC (Zhong et al., 2018; Hoyle et al.,
2011). Map of pH (Ballabio et al., 2019) since soil pH directly impacts microbial activities that influence the
turnover of soil organic matter, and consequently negatively correlates with SOC (Malik et al., 2018). Furthermore,
map of available water capacity (Ballabio et al., 2016) as this soil properties is another interactive factor with SOC
through plant productivity and soil texture (Burke et al., 1989; Yu et al., 2021). Soil erosion is also a key factor in
the SOC cycle ( Li et al., 2019), which was added through the soil erosion map of Europe (Borrelli et al., 2018).

### 2.3 Boosted Regression Trees

Developed by Friedman et al. (2000), BRT is a tree-based algorithm that applies boosting method to improve
accuracy. Boosting method relies on combining several approximate prediction models rather than obtaining one
single highly accurate one (Schapire, 2003). Thus, the decision trees are grown sequentially so that each decision
tree predicts the residual of the previous one. Consequently, the number of trees influences the performance of the
algorithm and requires tuning. However, to incorporate randomness into the model and subsequently increase the
robustness of performance, the trees are grown on a randomly selected data subset with no replacement (Friedman,
2002). The size of this subset is controlled by a parameter known as a bag fraction. Furthermore, the contribution
of each new tree to the final model is regularised by learning rate, also known as shrinkage(Friedman et al., 2009).
Finally, the number of splits in each tree that divides the response variable into subsets is optimised by interaction
depth. The BRT model was built in R using the "gbm" package (Greenwell et al., 2019).

### 2.4 Random Forest

Similar to BRT, RF is another tree-based algorithm. RF uses bootstrap sampling of the dataset for growing a
decision tree. Subsequently, by aggregating the results of a large number of decision trees, the bias and variance
of the final model can be reduced (Breiman, 1999). The method of bootstrapping in conjunction with aggregating,
known as bagging, increases the robustness and stability of RF. However, the trees from different bootstraps may
form a similar structure if all covariates participate in a split of each node. Thus, the variance cannot be reduced
optimally through the bagging process (Kuhn and Johnson, 2013). In order to avoid this tree correlation, a random
subset of predictors is selected at each split. The parameter $m_{try}$ defines the number of predictors included in this
subset and should be tuned (Kuhn and Johnson, 2013). The RF algorithm was implemented by the "Ranger"
package (Wright and Ziegler, 2017) in R.

### 2.5 Support Vector Regression

SVR is a form of support vector machine adopted for regression. From all possible solutions, i.e. estimation
function, for the problem, SVR tries to obtain an estimation function with the maximum $\varepsilon$ error while minimising
model complexity (Smola and Schölkopf, 2004). Thus, a symmetrical tolerance threshold, $\varepsilon$-insensitivity zone, is
created around the estimation function within which the vectors are not penalised (Awad and Khanna, 2015).
However, the vectors that lie on the boundary of the $\varepsilon$-insensitivity zone are called support vectors. Therefore, $\varepsilon$
is an optimisable parameter that controls the width of $\varepsilon$-insensitivity, alters the model complexity and impacts the
number of support vectors inversely (Cherkassky and Ma, 2004). Moreover, the trade-off between model
complexity and tolerance of $\varepsilon$ deviation is controlled by a parameter named C (Smola and Schölkopf, 2004;





Cherkassky and Ma, 2004). Optimising the C parameter has a crucial impact on SVR performance since a high C
can lead to overfitting, while a low C can cause under fitting (Kuhn and Johnson, 2013). The use of kernel functions
makes SVR a powerful tool for nonlinear problems. By implementing these functions, SVR can map data space
from its original dimension to a higher dimensional space where a nonlinear problem can be solved linearly. In
this study, the Radial Basis Function (RBF) kernel was used with gamma as its tuneable parameter. This parameter
affects the generalisation performance of SVR by controlling the influence of support vectors inversely (Battineni
et al., 2019). SVR was implemented from the package e1071 in R (Hornik et al., 2021).
**2.6 Performance evaluation**
When training a predictive model, it is important to evaluate its generalisation performance on unseen data of the
same type (Hawkins et al., 2003). However, as the number of available samples is usually a limiting factor, the
evaluation process is often done by randomly splitting the available dataset into training and testing sets multiple
times, i.e. cross-validation (CV). Although this process is effective, it is not entirely immune from biased
estimation of error. However, to ensure that the estimated error in model evaluation is as unbiased as possible,
every model training step should be performed within the CV. This includes finding the best parameter sets for the
chosen algorithm (Varma and Simon, 2006). Thus, the algorithms in this study were applied on a stratified nested
CV.
First, to ensure that the SOC distribution was represented in the CV scheme, Germany was divided using a 100x100
km INSPIRE grid into 50 strata. Random samples from each stratum were then taken and compiled into a fold.
This procedure was continued to create five folds and was repeated five times, forming the outer loop of CV used
for model evaluation. Large distance between neighboring samples, 8120 m on average, prevents train and test
data from being spatially autocorrelated. Since the aim was to tune the parameters of the algorithms, the training
set of the outer loop of CV was nested, creating five folds as the inner loop on which the parameter tuning was
performed. To evaluate the performance of algorithms, root-mean-squared error (RMSE), Eq. 1, mean absolute
error (MAE), Eq. 2, and mean absolute percentage error (MAPE), Eq. 3, were used.
$RMSE = \sqrt{\frac{1}{n}\sum_{i=1}^{n}(P_i - O_i)^2}$ (1)
$MAE = \frac{1}{n}\sum_{i=1}^{n}(P_i - O_i)$ (2)
$MAPE = \frac{1}{n}\sum_{i=1}^{n}\left|\frac{P_i - O_i}{O_i}\right| \times 100$ (3)
Where $n$ is the number of samples, $P_i$ and $O_i$ are the predicted and observed values, respectively.
**2.6.1 Parameter tuning**
As mentioned in Sect.1, choosing a suitable strategy for parameter tuning is a crucial step in machine learning
particularly for comparing the algorithms. Therefore, two strategies were applied depending on the algorithm: 1)
a grid search for RF and 2) optimisation with the DE algorithm for BRT and SVR. The first strategy was an
exhaustive search over a defined space consisting of lower bound, upper bound and $n$ steps in between for the
target parameter. Therefore, the target parameters in this strategy should be discrete or discretised beforehand if
they are continuous (Probst et al., 2019). This strategy was applied to RF since the tuning parameter is discrete.
However, the second strategy is a stochastic approach of searching over a continuous space in order to solve an



optimisation problem (Qin et al., 2009) and is described in more detail by Storn & Price (1997). Therefore, SVR
and BRT were optimised by this strategy as they have continuous parameters. For the optimisation task in the
present study, the R package "DEoptim" was applied (Peterson et al., 2021). Table S1 shows the parameters and
their tuning range for each algorithm.

**2.6.2 Variable importance**

Variable importance was assessed by permutation (Ließ et al., 2021). Therefore, each covariate in the test set was
shuffled 10 times and on each occasion the trained model corresponding to that test set was applied. The population
of RMSE was averaged and its relative change to the RMSE of the original test set was calculated. Thus, the
variable importance of each covariate in terms of percentage relative change in RMSE was obtained.

**2.7 Modelling approaches**

Three approaches were designed to test the performance of the algorithms. The models were built based on nested
CV, while the train and test sets remained identical for the three algorithms to make the results comparable. The
first approach (AP1) only used the SOC content from the German Agricultural Soil Inventory and corresponding
values from the covariates were used to build the models. Thus, the dataset was cross-validated and used by BRT,
RF and SVR to predict the SOC content of German agricultural soils. The results of this approach served as a
baseline on which the model improvement for each algorithm in other approaches was assessed.
Due to the high variability of SOC in agricultural soils of Germany, two separate models for organic and mineral
soils was developed and tested to identify whether it could improve model performance.  Accordingly, the German
Agricultural Soil Inventory was subdivided by the threshold 87 g kg$^{-1}$ into mineral and organic soils and then were
used to train separate models. This approach was named AP2. The same nested CV procedure was applied for both
data subsets. The results of BRT, RF and SVR were compared to identify which one had better performance under
mineral and organic soils separately. Finally, each algorithm's predicted SOC from two separate models was
combined, and the error metrics were calculated for the full data set to identify the impact of AP2 on model
performance. The CV folds for this procedure match the one from the AP1 models.
The impact of enlarging the training set on model performance was examined for both AP1 and AP2 approaches.
Thus, 1223 depth-extrapolated samples of the LUCAS data were added to the training sets of AP1 and named
AP1L. Moreover, the same threshold (87 g kg$^{-1}$) was used to subdivide this dataset and each soil class was included
to the training set of the corresponding soil class of AP2 and named AP2L. The test sets of the CV procedure
remained the same.

**3 Results and Discussion**

**3.1 Comparison of algorithms on the data from the German Agricultural Soil Inventory**

The range of SOC content of topsoil for the German Agricultural Soil Inventory dataset was 4 g kg$^{-1}$ to 480 g kg$^{-1}$
, with a mean of 27 g kg$^{-1}$ and median of 16 g kg$^{-1}$. Figure 1 shows the spatial distribution of the implemented
data. The RMSE and MAPE indicate that SVR had a better general performance than the two other algorithms
(Fig. 2). In this respect, the RMSE of SVR was 5% lower than that from RF and 4% lower than that from BRT.
Furthermore, its MAPE was 3% and 7% lower than that from RF and BRT respectively. However, despite the
difference in overall performance, the spatial distribution of relative residuals indicated that all three algorithms
were less accurate in the north of Germany compared with the centre and south of the country (Fig. 3A). This can





be explained by the characteristics of this region and its higher SOC variability. The northern part of Germany is
a lowland dominated by sandy soil texture from pleistoceen sedimentation with geomorphological structures such
as ground moraines, terminal moraines and aprons (Roßkopf et al., 2015). Despite general geomorphological and
pedological similarities throughout the region, 1) organic soils in Germany are mainly located in the north and 2)
mineral soils with the lowest and the highest SOC content are also located in the northeast and northwest
respectively. Therefore, this region has the highest SOC range on agricultural soils.

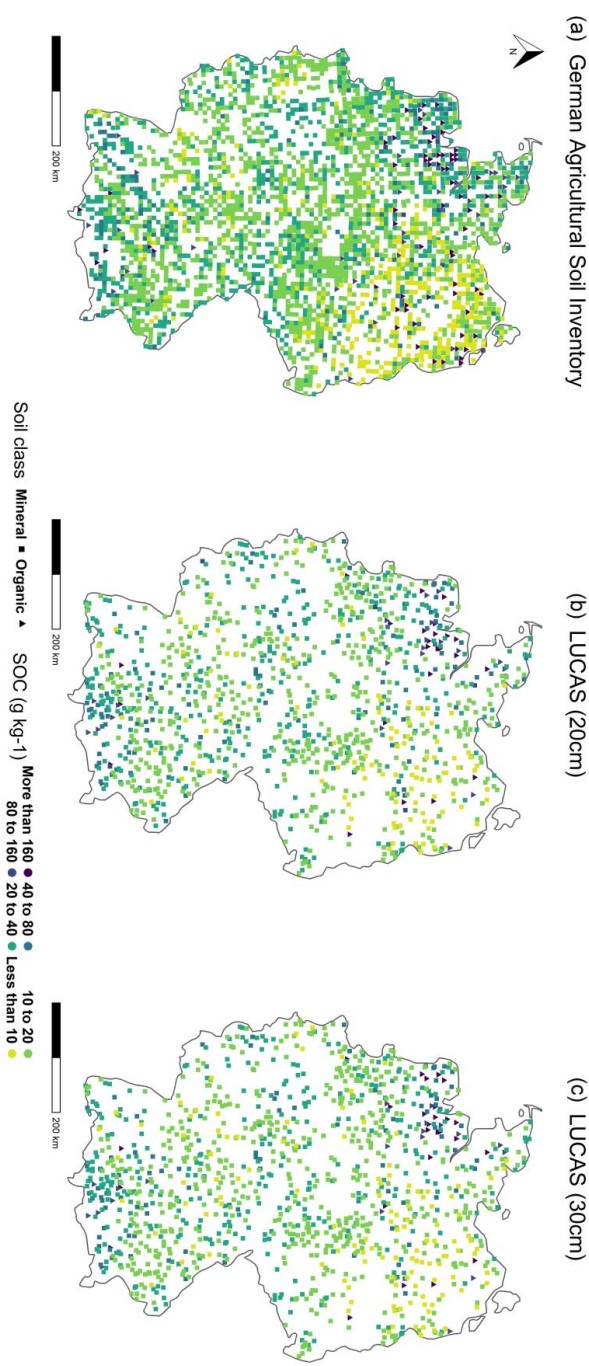


**Figure 1: Soil organic carbon content in topsoil of two soil inventories. A) German Agricultural Soil Inventory (0-30 cm), B) LUCAS at its original sampling depth (0-20 cm), C) LUCAS after depth extrapolation (0-30 cm)**

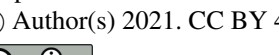



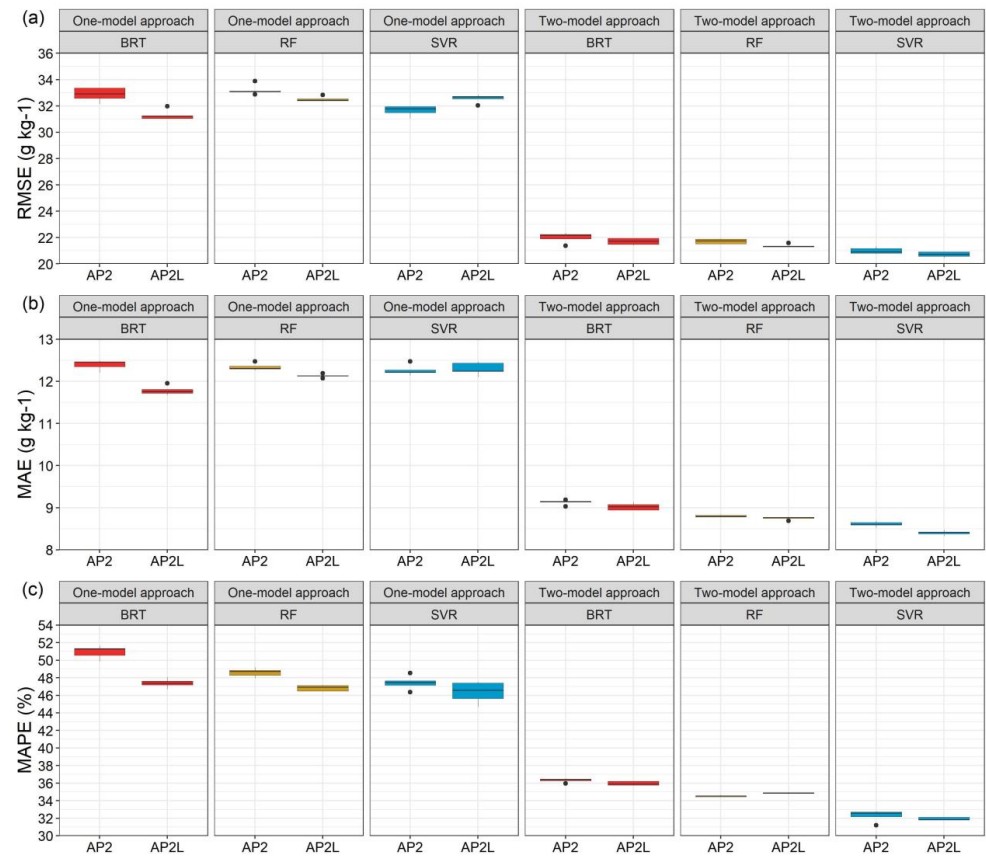


**Figure 2: Performance indicators of the three algorithms. One-model approach (Without LUCAS data AP1 and with LUCAS data AP1L) versus the two-model approach (AP2 and AP2L) for A) RMSE (g kg⁻¹), B) MAE (g kg⁻¹) and C) MAPE (%). Please note that the y-axis is shortened for better visibility and does not display a zero.**

Consequently, the variable importance (Fig. 4A) indicated that the map of organic soils contains the highest
available information among all covariates for the algorithms. The value for variable importance for this covariate
was 65% in SVR, 72% in RF and 84% in BRT. These values firstly show the crucial role of the map of organic
soils for the algorithms in explaining the variability of SOC and, secondly, how BRT mainly relies on the map of
organic soils to predict SOC compared with SVR. Despite the importance of the organic soil map, the scatterplots
(Fig. 5A) show that all three algorithms underpredicted the SOC of the organic soils and had similar
heteroscedasticity patterns in their residuals. Thus, while most residuals from mineral soils followed the 1:1 line,
they became more scattered in soils with a higher SOC content. The underprediction of SOC in organic soils can
be explained by their low sample size, resulting in a dataset with a high SOC range and a unimodal distribution
that leaves these soils in the tail. Consequently, the organic soils were underrepresented and the results were
systematically pulled towards mineral soils, regardless of the choice of algorithm. Different studies have shown
that predicting soil properties with mineral and organic soils combined can lead to underprediction or
overprediction of one soil class, depending on the distribution of the dataset (Brogniez et al., 2015; Guio Blanco
et al., 2018; Mulder et al., 2016).



Although the map of organic soils was able to distinguish between the two soil classes, i.e. between mineral and
organic soil, it could not separate the mineral soils with a low SOC content in the northeast from those with a high
SOC content in the northwest. The spatial distribution of the residuals (Fig. 6A) shows that SVR and BRT
generally underpredicted the mineral soils in the northwest part of Germany, while RF overpredicted them.
Furthermore, unlike RF and SVR, BRT distinctively overpredicted SOC of the north-east's mineral soils with the
lowest SOC content ($<10$ g kg$^{-1}$). This result indicates that the algorithms differed in their performance in mineral
soils. This difference was mainly due to the information they obtained from land use. As the second most important
covariate for all three algorithms (Fig. 4 A), the value for variable importance for this covariate was 22% in SVR,
but just 11% in RF and 9% in BRT. Thus, SVR exploits more information from this covariate than RF and
particularly BRT. Land use is one of the main drivers of SOC variability on a national scale due to the higher SOC
content in grasslands than in croplands (Poeplau et al., 2020). Therefore, this covariate was able to differentiate
between the soils of the northeast, which are under cropland, and those in the northwest as they are more under
grassland. Consequently, the reliance of BRT on the map of organic soils at the cost of land use could explain why
this algorithm overpredicted SOC in croplands in the northeast.






**Figure 3: Spatial distribution of relative residuals. A) AP1 approach, B) AP1L approach, C) AP2 approach and D)**
**AP2L approach.**





**Figure 4: Variable importance in terms of average relative change (%) in RMSE. A) AP1, B) mineral soil subset of AP2 and C) organic soil subset of AP2. The full name for each abbreviation is presented in Table S3.**



### 3.2 Enlarging the dataset with additional soil inventories


A larger soil dataset may provide additional information and consequently improve model performance. This
possibility was explored in the AP1L approach with adding the LUCAS data. The SOC content of LUCAS data at
its original depth ranged from 4 g kg$^{-1}$ to 500 g kg$^{-1}$ with a mean of 30 g kg$^{-1}$ and a median of 18 g kg$^{-1}$. After
extrapolating the depth to 30 cm, the new range was from 5 g kg$^{-1}$ to 512 g kg$^{-1}$ with a mean of 28 g kg$^{-1}$ and a
median of 17 g kg$^{-1}$. The spatial distribution of LUCAS data at their original and extrapolated depth is shown in
Figure 1.
A statistical test was performed on the residuals of models built on LUCAS data with the original and extrapolated
depths. That was done to identify whether extrapolating the depth of LUCAS data to that of the German
Agricultural Soil Inventory would significantly affect model performance after their inclusion in the training set.
With the Shapiro-Wilk test rejecting the normality assumption of residuals of all corresponding algorithms at 20
cm and 30 cm, the non-parametric Kruskal-Wallis test showed no significant difference between the residuals at
both depths. Thus, the extrapolation of the soil depth had no significant impact on the data quality to regionalize
SOC. As a result, any further change in the performance of the algorithms after adding LUCAS data was due to
the training set being enlarged. The result of the algorithms at both depths can be found in the supplementary
information (Fig. S1).
After enlarging the training set from 2278 to 3501 sampling points, BRT obtained the lowest RMSE and MAE
among the algorithms (Fig. 2). A comparison of the error metrics of corresponding algorithms from the AP1
approach with those from the AP1L approach showed that BRT had the highest error reduction at 7% in MAPE
and 5% in RMSE and MAE. Furthermore, although the error metrics of RF did not improve as much as those of
BRT, additional training points were still beneficial for this algorithm. However, SVR did not follow any
systematic change under the AP1L. Despite a 2% decrease in MAPE, RMSE increased by 3% and MAE remained
unchanged. To explore the potential explanation for this behaviour by SVR, the residuals of mineral soils were
separated from those of organic soils. Additional samples reduced the RMSE in mineral soils for all algorithms
between 9% and 13%. However, this error increased by 9% in the organic subset for SVR, while it increased by
just 1% for RF and even decreased by 1% for BRT. This indicated that enlarging the training set by data with
similar characteristics had a greater influence on systematic error of the underrepresented soil class in SVR. This
influence is understandable when considering the higher optimised $\varepsilon$ in the AP1L approach compared with that of
the AP1 approach. The higher value of $\varepsilon$ means that the hyperplane for the training set is less complex (Cherkassky
and Ma, 2004) and more suitable for predicting most soil samples, i.e. mineral soils. Thus, when this hyperplane
was fitted to the test set identical to the AP1, the generalisation performance was hindered because it could not
capture the variability of samples with higher SOC values, i.e. organic soils.
Further evaluation revealed that regardless of the change in error metrics, the relative residuals of the three
algorithms had a similar spatial pattern to their counterpart from the AP1. Thus, they all showed lower accuracy
in the northern region of Germany for similar reasons (Fig. 3B). Moreover, the scatterplots had a similar pattern
with underpredicted organic soils (Fig. 5B). This confirms that when organic soils are modelled with mineral soils,
enlarging the training set does not provide enough information for BRT or RF to capture the high variability of
SOC, particularly in the north of Germany.




**Figure 5: Scatterplot of residuals. A) AP1 approach and mineral and organic soils of AP2 and B) AP1L approach and mineral and organic soils of AP2L.**





Figure 6: Spatial distribution of residuals. A) AP1 approach, B) AP1L approach, C) AP2 approach and D) AP2L approach.





### 3.3 Subdividing soil inventories into mineral and organic subsets


As presented in the sections above, the modelling of SOC content when mineral and organic soils were combined
led to a systematic underprediction of soils with higher SOC values by all three algorithms, regardless of the
number of training samples. Therefore, by implementing the AP2 approach with two models one for mineral soils
and one for organic soils, a noticeable improvement in the performance of all algorithms was observed, with SVR
showing the best error metrics (Fig. 2). This meant 34% lower RMSE, 30% lower MAE, and 32% lower MAPE
than when this algorithm was trained under the AP1 approach with one model for all soils. As the high variability
of SOC was initially hard to capture, the subdivision of the dataset provided a range that better represented each
soil class. This was particularly beneficial for mineral soils (ranging from 4 g kg$^{-1}$ to 85 g kg$^{-1}$) since the number
of samples did not reduce drastically (only by 99 samples). Thus, the algorithms could better capture the relation
between SOC and covariates. Consequently, the overall performance improved when the underrepresented soil
class was modelled separately. This is in line with the study of Rawlins et al. (2009) which recommends the
separate modelling of mineral and organic soils.
Nonetheless, following the AP2L approach with additional data, the RMSE and MAPE of the algorithms improved
by less than 2% compared with AP2. However, the greatest change was observed in the MAE of SVR with a 2%
improvement. Therefore, additional training samples did not considerably influence the performance since the
majority of these samples were in mineral soils, while the limiting factor was the high variability of organic soils
combined with its low number of samples. Nevertheless, an improvement was noted in relation to the all error
metrics of SVR in the AP2L approach. This was in contrast to when the training set was enlarged without
subdividing the data, i.e. AP1L. Therefore, it further confirmed that it is more important for SVR than BRT and
RF to model the soil classes separately when its training set is enlarged by datasets with similar characteristics.
Furthermore, the improvement of the algorithms in AP2 and AP2L was particularly noticeable in their relative
residuals. By comparing these results with those from AP1 and AP1L, it was evident that the greatest improvement
was observed in the northern region and the spatial distribution of relative residuals was more homogenous
throughout the country for all algorithms, but particularly for RF and SVR (Fig. 3 C and D). This is understandable
since by subdividing the data, the algorithms can no longer exploit any information from the map of organic soil
for spatial variability of SOC in mineral soils. Thus, they obtain information from other covariates for this soil
class (Fig. 4 B). Although land use and total nitrogen were still among the most important variables for the
algorithms in mineral soils, the importance of the predictors representing the SCORPAN C and P factors increased
in the absence of a soil organic map. This could be expected because the north-east of Germany, for example, has
continental climate (Roßkopf et al., 2015) and young moraine landscapes, while the north-west has a more oceanic
climate (Roßkopf et al., 2015) with old moraine landscapes.
It is unsurprising that all the algorithms still relied on the map of organic soil to explain SOC in organic soil class.
However, while SVR and RF still obtained information from other covariates, the value for variable importance
of this map alone is 93% in BRT (Fig. 4 C). That makes this algorithm prone to greater errors, as can be seen in
its error metrics (Table S2). Similar to mineral soils, the order of covariates was different between the algorithms
in organic soils. In other words, in AP1 the three algorithms obtained almost all information from the map of
organic soil, land-use and total nitrogen with similar order. In contrast, after subdividing the data, the algorithms
differentiated from each other by the order of covariates in their variable importance (Figure 4).





A comparison of the error metrics of each soil class in AP2 with its counterpart in AP2L revealed that the additional
1177 samples had a minor influence on the performance (from zero to a maximum of 2%) of the algorithms in
mineral soils (Table S2). These results indicated that the German Agricultural Soil Inventory offers a good
representation of the spatial variability of SOC in mineral soil under agricultural use throughout the country and
including more sample points do not provide additional information about SOC variability in this soil class.
However, 46 additional organic soil samples from the LUCAS dataset improved MAPE and MAE by 12% and
6% for SVR, by 10%, and 4% for RF, and by 7% and 2% for BRT, respectively, but the RMSE of the three
algorithms was improved by less than 2%. Thus, additional organic samples mainly influenced the average
magnitude of the error. This could be explained by organic soils having a wide range of SOC and the number of
samples was limited. Thus, the addition of LUCAS data to the training set offered the algorithms more information
about spatial variability of SOC in this soil class. Despite this limitation, SVR had the best overall performance
among the algorithms in AP2 and AP2L. It should be noted that training samples must span the complexity of the
parameter space in order for the model to be able to effectively match the training data and to generalize unseen
data. Small sample size can therefore negatively influence the predictive power of the algorithms. This complexity
can be addressed by structural risk minimisation (SRM) (Al-Anazi and Gates, 2012). Implementation of SRM
makes SVR capable of performing well in such datasets. Other studies have compared the performance of
algorithms on different sample sizes for predicting soil properties and shown that SVR is one of the best choices,
if not the best, when the number of samples is a limiting factor (Al-Anazi and Gates, 2012; Khaledian and Miller,
2020). In contrast, in a study by Zhou et al. (2021), 150 samples with different sets of covariates at different
resolutions were used to compare RF, BRT and SVR to predict SOC content in Switzerland. Their results showed
that the best-performing algorithm varied depending on the resolution and covariates. However, the best
performance throughout all scenarios was obtained by BRT. The discrepancy between their results and the results
of the present study may be due to the parameter-tuning method of the algorithms, as they only used grid search,
or other factors, including the spatial distribution of samples or the chosen set of covariates.
**4 Conclusions**
The three most commonly used algorithms in DSM were implemented to predict the SOC content of German
agricultural soils under different approaches. Suitable tuning strategies for each algorithm ensured optimum
parameter tuning and made their performance truly comparable. Machine learning algorithms was shown to be
powerful in modelling SOC on a national scale. However, the study showed that separate modelling of mineral
and organic soils was a better approach for modelling SOC compared to using one model. Thus, this approach has
priority to the choice of algorithm and number of training samples. We recommend this approach to be further
tested in countries and regions that cover both of these soil classes. Nonetheless, SVR had better performance than
RF and BRT except when the number of samples in training was increased. This was disadvantageous for SVR
and advantageous for BRT unless mineral and organic soils were modelled separately. Therefore, this approach
should be done with consideration of the algorithm and the characteristics of the data. Furthermore, better
performance of SVR over RF and BRT was particularly highlighted when predicting SOC in organic soils. Thus,
this algorithm should therefore be taken into greater account in DSM when the number of samples is limited.



**Data availability**

The soil data used in this study are publicly available via: https://doi.org/10.3220/DATA20200203151139 and
https://esdac.jrc.ec.europa.eu/content/lucas-2009-topsoil-data

**Author contribution**

AS and AD conceptualised and developed the methodology of the presented work, with input from ML.AS
gathered the predictors with contribution from AD. AS executed programming, testing of existing code
components, formal analysis and visualization. AG contributed to programming. The preparation of the paper was
done by all authors.

**Competing interests**

The authors declare that they have no conflict of interest except author AD is a member of the editorial board of
the journal.

**Acknowledgement**

This work is part of the SoilSpace3D-DE project. The LUCAS topsoil dataset used in this work was made available
by the European Commission through the European Soil Data Centre managed by the Joint Research Centre (JRC),
http://esdac.jrc.ec.europa.eu/".



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
