# Peer review of "Spatial prediction of organic carbon in German agricultural topsoil"

_SOIL, 2021_

## Author Response (AR1)

Please note that the line numbers in our answers are referring to the updated manuscript.

**Anonymous Referee #1:**

Overall this is a very interesting paper in which a fair comparison as regards different DSM approaches has been made across Germany, including the effect of a data-size extension (after combining 2 databases) and whether mineral and organic soils should be treated separately (by creating two different models).

In addition, the paper is well structured and writing, though some minor spellings and grammar improvements are possible (please note that I only focused on the language in the first couple of pages, but I'm convinced that the entire paper could benefit from some slight language polishing)

Nevertheless, I believe that this paper may require some major revisions based on following comments:

**Answer:**

- Thank you very much for taking the time to review our manuscript and your constructive comments.
- The manuscript is revised for grammar and spelling improvement.

**Main Suggestions/Remarks**

I.1 This research considers agricultural soils, including both grassland and cropland, and as such I have some serious concerns as regards the presented (0-20 to 0-30) depth interpolation approach in order to match both databases (P. 3), which seems to be based on a (first order?) linear function depending on the soil class. However, in my opinion this analysis should be carried out per land use – soil type combination, because the depth distribution in cropland topsoil is remarkably different to that in grassland topsoil (i.e. more or less a Cte value versus exponential decline, respectively). Hence, I would like to ask the authors to carry-out this analysis again per land use soil – type. Moreover, only a (general / average?) slope parameter value has been given (in L115 – 116), and as such I would like to ask the authors to provide the readers with a much more detailed picture on the different slope parameter values obtained depending on the land uses (and soil types) setting. This can be done in a tabular format (in annex) by presenting the slope parameter (+/- the associated SE) for each land use and soil type combination - or - in a graphical format showing the distribution of slope parameter values per land use type.

**Answer:**

- We agree and acknowledge the argument about depth extrapolation based on land use - soil type combination. However, the main difference in depth distribution between cropland and grassland is within 0-20 cm and this difference is lower in 20-30 cm. Nevertheless, we compared the suggested and the implemented extrapolation approaches using Kruskal-Wallis Test and there is no statistically significant difference between the two. Therefore, we believe that depth extrapolation using either of the methods do not have significant impact on the outcome of the models.

- We agree and provide more transparency by including the plots for cropland, grassland and different soil types in the supplement under Figure S2. Moreover, the equations that were used for extrapolation is included in the manuscript (Eq. 1 and Eq. 2) in section 2.1 (L120, L121)

I.2. From section 2.2 I can see that a wide range of covariates has been considered. However, I was wondering whether the authors did carry-out any multi-collinearity analysis in order to identify those who may be too strongly correlated (e.g. r > 0.9). Subsequently, I was wondering what they have done to solve this potential issue?

**Answer:**

- We agree that multicollinearity can be problematic for interpreting the importance of covariates that are multicollinear. However, as the main aim of the study is model performance (L98-103), multicollinearity is not such a concern. This is supported by "applied linear statistical models" by Kunter et al (2005) pp:283.;
  - "The fact that some or all predictor variables are correlated among themselves does not, in general, inhibit our ability to obtain a good fit nor does it tend to affect inferences about mean responses or predictions of new observations, provided these inferences are made within the region of observations."
- Furthermore, the variables that were the focus of the discussion are either categorical (such as map of organic soils and land use) or do not have strong collinearity (>0.9) with any other covariates (such as total nitrogen). The only variables with collinearity are clay and available water capacity.

I.3 The model performance evaluation indicators (section 2.6) are all quite similar and have a particular focus on "random error". Hence, I would like to suggest to include some others that could provide the readers with some information as regards the (%) bias. In addition, within 'the spirit' of SVR I think that including also a model performance evaluation indicator that also takes into account the concept of 'model complexity' could be an interesting add-on here. (I know that in the context of this kind of model this can be interpreted quite widely and may include a penalization term that depends on the number of parameters (like AIC and BIC) or the complexity of the trees / nodes, ect....)

**Answer:**

- We thank you for pointing out this shortcoming. We included AIC, BIC and %Bias to the Table S2.
- Moreover, the equations to calculate AIC, BIC and %Bias is included as Eq. 6, Eq. 7 and Eq. 8 respectively to section 2.6 "Performance evaluation" (L231, L232, L233).

I.4 I think that the "Results and Discussion" section requires some clarifications as regards the structure. In essence, I would like to suggest to add a short intro-paragraph explaining briefly the logic behind the structure (and clarifying as such the meaning of AP1, AP1L, AP2 and AP2L). Moreover, in the (bold) headings of the separate sub-section you could add the corresponding abbreviation in brackets to it at the end as well as give a short statement at the start of the section which case you're going to consider (actually, similarly to what have been done in L318). In addition, I believe that in some cases a bigger effort could be made to discuss

the regional differences (in results obtained by applying the different approaches). In that respect I would like to suggest the authors to provide the readers with a relative residual map with annotation of + or – in order to be able to interpret the under / over predictions patterns in a spatial explicit way (I think this will have more value that the maps in fig. 3 and 6).

**Answer:**

- We agree that the structure requires more clarification. Therefore, we have fully adapted section 2.7 "Modeling approaches" (L256-272).
- Moreover, we included Table 1 to the section 2.7 (L258) for further clarification of our modeling approaches.
- The corresponding abbreviations (AP1, AP1L, AP2, AP2L) are also included to the headings of the sub-sections in the "Results and Discussion" section (L274, L333, L378).
- L277-278 are added to section 3.1 for further clarification.
- We appreciate your comment regarding the relative residual figure with + and – annotation. However, the pattern of suggested figure does not differ from the ones that are already provided. Nonetheless, the requested relative residual figure is also included to the supplement as Figure S4.

I.5 I believe that 'the main message' should be highlighted more, i.e. the fact that creating 2 separate models (one for mineral soils and one for organic soils) is much more important than the choice of the type of model (at least those considered here) and/or the suggested data-size extension. Please make sure that this is highlighted in the discussion and the conclusions sections. In that respect I think that some small additional analysis could be useful, for example a table / figure showing the potential model improvement (e.g. average RMSE decrease – or any other model performance indicator – see comment I.3.) due to this 3 factors (i.e. model separations (org vs. mineral), type of model, data extension). I think that this can be calculated rather easily from the information given in figure 2.

**Answer:**

Thank you for your suggestion. We agree that the main message of the paper is about optimizing SOC mapping by testing different approaches and not only comparing different models. Therefore:

- The title of the paper is adapted to: Spatial prediction of organic carbon in German agricultural topsoil using machine learning algorithms
- The abstract is adapted to highlight the two methods (L12-15).
- L443-451 are included to further highlight the main message in the "Results and Discussion" section.
- Table 2 is included to the manuscript (L442).
- However, the main message was pointed out in conclusion (L466-469).
- Moreover, percent change in error metrics between modeling approaches, model types and data extension are included to supplementary materials under Table S3.

I.6 As I understood (from reading section 2.2) that all the covariates are represented by a spatially continuous map, I was wondering whether it could also be an option to provide the readers with one spatially continuous predicted SOC map, for example created by applying 'the best model' (AP2L?) on the various covariate maps. I think this could be useful in order to obtain a more detailed interpretation of the results taking into account regional differences depending on various environmental settings.

**Answer:**

- We appreciate the suggestion and included the spatial prediction of SOC to the supplement as Figure S5.
- L452-461 are included.

**II. Specific Suggestion/Remarks**

L9-10: "to influence climate change and mitigation" is a somewhat strange formulation. (I guess this should have been "to influence and mitigate climate change"?) Please rephrase.

**Answer:**

- It is rephrased to: has the potential to influence and mitigate climate change (L9-10)

L 15: define topsoil, (e.g. add '(0-30cm)')

**Answer:**

- The definition (0-30 cm) is included (L14)

L 32 – 37: you make several references to Meersmans et al 2012 but in your reference list there is 2012a and 2012b, so please specify "a" of "b" here.

**Answer:**

- The references are corrected by including "a" and "b" to the corresponding ones (L32, L35, L37)

L 46: "at a different scale" is a somewhat strange formulation. (I guess this should have been "at different scales' or 'across different scales") Please rephrase or delete.

**Answer:**

- "at a different scale" is deleted and the sentence is rephrased accordingly (L45-48).

L54: I suggest replacing "SOC inventory" by "SOC monitoring" because you make reference to the periodic character of it.

**Answer:**

- "SOC inventory" is replaced by "SOC monitoring" (L54)

L57: "with a sampling depth down to 100 cm" is a somewhat strange formulation. A more common way to say this could be "considering a sampling depth of 1m" or "considering a reference depth of 1m".

**Answer:**

- It is rephrased to "considering a sampling depth of 1m" (L57)

L61: What do you mean with "complete"? Is this a good spatial distribution? Please clarify.

**Answer:**

- The sentence is rephrased for further clarification (L60-63).

L73: add a space between "(2017)" and "concluded".

**Answer:**

- It is included (L73)

L 81: What do you mean with disparity? (Do you mean "sample design"? Or "spatial distribution"?) Please clarify.

**Answer:**

- It is replaced by "data density, quality, representativeness" (L82).

L 120-126: Why didn't you just use just the best quality product? Are all covariates resampled to a resolution of 100m? And if yes, why not use the any higher level of detail / precision if you have been provided with it anyway? Was this done in order to deal with some computation intensity issues?

**Answer:**

- All covariates were either rasterized or resampled to 100 m resolution. The 100 m resolution was chosen as a compromise between computing capacity, a useful resolution for quantitative applications on agricultural soils in Germany, and the input data.

L 137: What is the (initial) resolution of this DEM? Was this layers also resampled to 100m (see previous comment)? And if so, was this done before or after deriving the related co-variates (such as slope, curvature ect…). Please be more clear / specific about the exact methodological approach followed here.

**Answer:**

- The sentences are revised for further clarification (L140-142).

L145-147: please make a reference to the source of this map.

**Answer:**

- The reference is included (L150).

L 164: What kind erosion map has been considered? Is it a map highlighting water erosion and/or tillage erosion? Hence, please specify what kind of model has been considered to generate this map (e.g. Is this map based on RUSLE or WatemSedem)? I'm also wondering whether it was really required to add this map, because you have already a lot of topographical related input variables which may provide you with similar info. (In that respect I like to reiterate my main comment I.2 – see above)

**Answer:**

- Explanation about the erosion map is included and the section is revised for further clarification (L169-172).
- There is no multicollinearity between this map and other covariates. Also, regardless of multicollinearity, it was expected that the obtained information from this covariate and the topographical ones would differs for different algorithms and under different approaches.

L 175: What kind of interaction depth did you consider?

**Answer:**

- The interaction depth was optimized between 1-5 and indicated in Table S1 of the supplementary materials.

L 188: Are you sure this needs to be "maximum error"? To me it sounds more logic to go for a model with "minimum error" but still with a limited model complexity.

**Answer:**

- Although intuitively we aim for minimum error, in the framework of SVR, certain level of error is tolerated during training phase to avoid overfitting (bias-variance tradeoff). Thus, SVR behaves similar to soft margin SVM. In this regard, the margin (maximum tolerated error) within which the error is not penalized is determined by epsilon.
- However, we acknowledge that the phrasing could be misleading thus it is rephrased to (L195-197):

o SVR tries to obtain an estimation function that has at most $\varepsilon$ deviation from the response values of the training data while minimising model complexity (Smola and Schölkopf, 2004).

L237: Can you clarify what you exactly mean with "shuffled 10 times". I guess this is a kind of random perturbation? (following a normal distribution?) Is it similar to what one will do in Monte Carlo?

**Answer:**

- To determine the importance of a given covariate, the values of that covariate in the test set was shuffled and the trained model was run on the test set to calculate the RMSE. This process was repeated 10 times and the mean of 10 RMSE was calculated. Comparing this RMSE with the obtained RMSE from the model when it was tested on the original test set would determine the importance of the given covariate.
- Section 2.6.2 is modified for further clarification of the method (L250-254).

L 265 / Figure 2: Please add subplot labels to fig2 (a1, a2, a3, ….. b1, b2,…) and make always reference to the specific subplots in the text so the reader know immediately which subplots needs to be considered / compared (and which one he / she can ignore).

**Answer:**

- Subplot labels included above the boxplots (1, 2, …).
- L349-350 are modified:
    - o BRT obtained the lowest RMSE (Fig. 2A1) and MAE among the algorithms (Fig. 2B1).
- L383 is modified:
    - o with SVR showing the best error metrics (Fig. 2A6, Fig. 2B6, Fig. 2C6).

Figure 2: Besides adding subplot labels (see comment just above this one), I think there is an error in the x-ax labeling, because in all cased it is either "AP2" or "AP2L", so there is no "AP1" or "AP1L" present, whereas I think that all the plots on the left-hand side of the figure (which are making reference to "one model approach") should have the labels "AP1" or "AP1L" (and not "AP2" or "AP2L" is currently the case). Right?

**Answer:**

- Labels are corrected.

Figure 5: Please add a regression line though these clouds of dots so one can evaluate a potential bias and /or over- /underprediction. (please note that this suggestion is related to my main comment I.3)

**Answer:**

Thank you very much for your suggestion. However, due to leverage effect of large values which can influence the regression, the regression line cannot be very helpful particularly in full dataset where we have many large values. Therefore, as we aim for more consistency between the results, we think including regression line is not helpful in this case.

**Anonymous Referee #2:**

In general, a well-written paper that presents soil organic carbon modelling at a national scale (Germany). The author focus on three aspects, namely the comparison of three machine learning models, expanding the national dataset with samples from a continental scale survey (LUCAS dataset) and how generating two separate models for mineral and organic soils affects the performance of such models.

**Answer:**

- Thank you very much for taking the time to review our manuscript and your constructive comments.

**General comments**

I have a problem with the way maps are presented. As far as I understand, the paper is a digital soil mapping (DSM) study but I do not see any maps with continuous predictions but just some points on a map. Or are those the areas corresponding to croplands? Please clarify. Second, you use a discrete colour map to show the results which do not allow the reader to see the spatial pattern of the predicted maps. You discuss the distribution of the residuals but a more detailed visual inspection of maps could be useful (which is common in DSM). For instance, Boosted Regression Tree (BRT) seems to mostly use categorical covariates (except for total nitrogen). How does that map look like?

**Answer:**

- We appreciate the suggestion and include the spatial prediction of SOC to the supplement as Figure S5.
- L452-461 are also included.
- The mentioned residual figures (Figures 3 and 6) show the relative error and residuals of models for sampling points in cropland and grassland and in both mineral and organic soils.
- Figure S1 is provided for better visualization of the covariates.

The largest difference can be seen when you split the dataset in mineral/organic. There is no doubt that the difference is significant. What about the rest of the comparisons? You use a Kruskal-Wallis to show that extrapolation in depth of the LUCAS dataset is valid but it is not clear if the main comparison (between three models according to the title) is significant.

**Answer:**

- This is a really good point. The problem with statistical test in this case is the discrepancy between the change in error metrics and the test results. As an example,

while Kruskal-Wallis test shows significant difference between all three algorithms in AP1 approach, the difference between the RMSE and MAE of BRT and RF is less than 1%. On the other hand, the same test indicates that the difference between BRT and RF is insignificant in mineral soils, yet the difference between their RMSE and MAE are 4.2% and 4.8% respectively. Furthermore, SVR and RF are significantly different in the same soil class while their RMSE is only 1.1% different. Beside the aforementioned problems, the statistical test is always insignificant in organic soils due to its low sample size. Therefore, such tests will not provide a satisfying answer whether the differences are significant and relevant and they very much depend on the sample size. Thus, the statistical test can be misleading in this case.

Perhaps the paper is focussing too much on the differences between models which is not very interesting. We have seen hundreds of papers comparing different models just to confirm that the "best model" depends on many factors. However, your results on modelling mineral and organic models separately seem interesting and perhaps focussing on that could benefit the community and the readers.

**Answer:**

We agree that the main message of the paper is about optimizing SOC mapping by testing different approaches and not only comparing different models. Therefore, we revised different parts of the paper:

- The title of the paper is adapted to stress more the particular challenges of modelling SOC due to the large value range stretching from mineral soils to few organic soils: Spatial prediction of organic carbon in German agricultural topsoil using machine learning algorithms
- The abstract is adapted to highlight the two methods (L12-15).
- L443-451 are included to further highlight the main message in the "Results and Discussion" section.
- Table 2 is included to the manuscript (L442).

How do you actually use two separate models (mineral/organic) in practice? In this approach, to make a SOC prediction you first need to decide which model to use. But to make that decision, you need to know the SOC concentration. This is an important point that should be discussed. For instance, how do we generate a national map in this particular study? Is your potential solution applicable to other countries?

**Answer:**

- That is a valid point. In this study the map of organic soil is used to separate the regions with organic soils from the ones with mineral soils. Thus, other data sources such as map of soil types or soil units are required to delineate organic and mineral soils. Then, each model was applied to its corresponding soil class. This method can be implemented where spatial distribution of organic soil is known.

- This point is also included for further clarification (L452-453).

I think a bit more discussion about the covariates could be useful. Many of the soil covariates used correspond to continental scale predictions (with significant uncertainty) which usually perform poorly at other scales (national). In addition to that, is interesting to see how just a few covariates are actually used by the models. Are we using too many useless covariates in DSM (studies with dozens of covariates)?

**Answer:**

- We acknowledge the potential limitations of using continental covariates for national prediction. However, most nationwide products are mainly derived from Soil Map of the Federal Republic of Germany (BUEK1000) by assigning the dominating soil type to each map legend unit. Thus, neglecting the others. Furthermore, a map such as the one in the study "Topsoil texture regionalization for agricultural soils in Germany–an iterative approach to advance model interpretation" by Gebauer et al. (2022) was not available at the time of study.
- This study cannot determine whether too many useless covariates are used in DSM or not. That would depend on various factors such as availability of covariates that can explain the variance of the target. As in our case, contribution of the covariates also depended on the modeling approach and the algorithm.

**Specific comments**

Section 2.6.1: I think the way parameter tuning is described is not correct. First, you mention that grid search parameters need to be discrete or discretised, which is not true. You can use continuous parameters without problem (e.g. [1.0, 0.1, 0.01]). Second, you used a DE algorithm for BRT since the parameters are continuous but `number of trees` and `interaction depth` are discrete. Based on your criteria, you couldn't use any of the strategies for BRT. A clarification is required.

**Answer:**

- Thank you for your comment and pointing out the shortcoming. The section 2.6.1 is fully revised (L237-248).

Figure 2: The limits of the whiskers and boxes sometimes represent different things depending on the library. Please add what they represent in the caption.

**Answer:**

- The explanation is included in the figure caption (L295).

---

## Author Response (AR2)

Please note that the line numbers in our answers are referring to the updated manuscript.

**Anonymous Referee #2:**

**General comments**

I still have a problem with the main map (Fig S5) being in the supplements instead of the main part (given the title of the manuscript). Also, at the end of the discussion, the authors mention that "all three algorithms showed a relatively similar distribution of SOC content across the country particularly in mineral soils." Is it an effect of the colour scale? I think using a log scale would better highlight the patterns. At the moment we can only see some blueish blobs with some lighter areas in the northern and southern extremes.

**Answer:**

- The map of SOC content is relocated to the main text under Figure 7.
- Thank you for the suggestion, the maps are log scaled and the color scale is changed to highlight the pattern better.
- L461-464 is now combined.

Also, at the end of the supplements, there is a very important Disclaimer. Instead of hiding it at the very end, I think it should be incorporated as part of the discussion since it is raising some important points.

**Answer:**

- Thank you for your recommendation. We included the disclaimer in the caption of Figure 7 with further explanation in the L469-476